# Shifting the Care of Type 2 Diabetes Mellitus from Hospital to Primary Health Care Institutions through an Educational Intervention for Health Care Professionals: An Example from Rural China

**DOI:** 10.3390/ijerph17062076

**Published:** 2020-03-20

**Authors:** Shaofan Chen, Dongfu Qian, Bo Burström

**Affiliations:** 1Health Outcomes and Economic Evaluation Research Group, Department of Learning, Informatics, Management and Ethics, Stockholm Centre for Healthcare Ethics, Karolinska Institutet, SE-17177 Stockholm, Sweden; 2Equity and Health Policy Research Group, Department of Global Public Health, Karolinska Institutet, SE-17177 Stockholm, Sweden; bo.burstrom@ki.se; 3School of Health Policy and Management, Nanjing Medical University, No. 101 Longmian Avenue, Nanjing 211166, China; 4Centre for Health Policy Studies, Nanjing Medical University, No. 101 Longmian Avenue, Nanjing 211166, China

**Keywords:** diabetes care, health care professionals, educational intervention, rural China

## Abstract

This study assessed the impact of an educational intervention on the knowledge, attitudes, and practice regarding Type 2 Diabetes Mellitus (T2DM) of Primary Health Care (PHC) professionals, as well as on the types of T2DM care services which they were able to provide. The intervention was carried out in collaboration with county hospitals. The study was conducted from 2015 to 2016 among 241 health care professionals in 18 township health centers and 55 village clinics in three counties in Jiangsu Province, randomly divided into an intervention group and a control group. Participants in the intervention group received professional skills training sessions and team communication and were involved in regular meetings. The control group followed the routine work plan. At one-year follow up, the diabetes knowledge score, practice score, and attitudes score were significantly higher in the intervention group than in the control group. A significantly higher proportion of health care professionals in the intervention group was able to provide services compared with the control group, for all types of services, except T2DM emergency treatment. The intervention among health care professionals in PHC had a positive impact on their professional diabetes skills, knowledge, attitudes, practices, and types of services they were able to provide, at one-year follow-up.

## 1. Introduction

The global prevalence of Type 2 Diabetes Mellitus (T2DM) is rapidly increasing as a result of fast urbanization, population ageing, and lifestyle changes [1]. In the last 30 years, the number of T2DM patients has more than doubled globally, making it one of the most important public health challenges to all nations [2]. T2DM is becoming a threat to people’s health in China as well. The overall prevalence of T2DM increased to 10.9% in 2013 [3,4]. The situation in rural China is even more difficult, with a faster increase rate of T2DM incidence than in urban areas, while the awareness, treatment, and control of diabetes remain lower in rural areas [5].

The previous diabetes care strategy before 2009 focused more on the costly hospital care than on primary care and self-management in rural China, and integration of care between hospitals and primary care services was uncommon [6]. Primary Health Care (PHC), however, has been proven to be the key setting for the effective management of chronic diseases, especially in low-income and middle-income countries [7,8]. In 2009, China launched a new round of health care reform aiming to offer universal coverage of essential health services for all Chinese citizens by the year 2020 [9]. As one of the most important components, PHC was strengthened, and the subsidies were increased sharply by the Chinese central government [10]. Meanwhile, policies were also implemented to support PHC [11]. In July 2019, a public health policy called “Healthy China 2030” was launched which is a new round of health care reform [11], focusing on the determinants of chronic disease care, such as environmental health, lifestyle, and health education [11].

Despite these opportunities for PHC development, challenges exist for shifting the care of chronic diseases from hospital to PHC, especially for the health care professionals in PHC in rural China. The current PHC institutions in rural China consist of township health centers and village clinics, and almost all of them are publicly owned [12]. However, the doctors and nurses there have a low level of training for chronic conditions, commonly in the lower level medical education [12]. China has a system of multi-tiered medical education to training physicians, including the full-time professional training program in medical university and the short or partial training in medical technical school or junior medical college [13]. The majority of health care professionals in rural China only receive a three-year medical education in medical technical school or junior medical college and lack access to updates in medical knowledge and skills [14]. Studies have shown that the professional ability in PHC needs to be improved [15]. The quality of PHC service in rural China has also not been well evaluated [12]. Hence, efforts are needed to develop a cooperation between PHC and higher level hospital care and to improve the professional skills and knowledge of health care professionals in PHC, in order to improve the care of patients with chronic disease. Chronic diseases-related programs which focus on training health care professionals in PHC have been conducted all around the world [16,17,18,19,20,21], while few studies have focused on China [22,23], particularly in rural areas [24].

More knowledge is needed of the vertical integration between hospital and PHC, and support from hospital-level to health care professionals in PHC. The present study reports on an educational intervention in collaboration between the county hospitals and local-level health care services directed to health care professionals in PHC institutions in rural China from the year 2015 to the year 2016 and its impact on professional knowledge on diabetes, attitudes and practices in interventions, and types of diabetes care services the professionals were able to provide.

## 2. Materials and Methods 

### 2.1. Study Design and Study Setting

The study is part of a research project on vertical integration strategies in health services for rural patients with chronic diseases (“Studying the Vertical Integration Strategy of Chronic Disease Service Based on Multiple Incentive Mechanism in Rural China”, ISRCTN13319989) [25], which focuses on optimizing the care of patients with chronic disease in three pilot counties in rural areas of Jiangsu province, through improved collaboration between the county-level hospital and PHC. Patients with T2DM and primary hypertension and health care professionals in PHC in rural areas received an educational intervention to shift the care from hospital to PHC services and to improve the vertical integration in health care for patients with these conditions [25]. The present study focuses on health care professionals and concentrates on T2DM.

Jiangsu Province, where the intervention was implemented, is commonly divided into three parts (north, middle, and south) according to geographic features and economic development [26]. We used simple random sampling to select one county from each part of Jiangsu Province. Huaiyin was selected from 37 counties in the north part. Jingjiang was selected from 11 counties in the middle part. In the south part, Gaochun was selected from 39 counties. Huaiyin consists of 14 townships [26]. Jingjiang and Gaochun have both eight townships [26]. In these three counties, 2–4 townships were selected as intervention areas by the local county-level Health and Family Planning Commission (HFPC). Subsequently, considering sociodemographic features, economic development situation, and medical care level, 2–4 comparable control townships were selected in the same county. In total, 18 township health centres (9 intervention, 9 control) were part of the study. Each of the townships represents a population of 30,000 to 100,000 on average.

### 2.2. Study Population

Health care professionals in PHC were invited to participate in the study. At baseline data collection, we selected our interviewees from both intervention and control areas, according to the inclusion criteria: should be frontline staff (not administrative or personnel staff); should be willing to take part into the data collection; should have worked in the current institutions for at least 2 years and have no plan to leave. Two questionnaires were provided to them, a professional knowledge questionnaire and a questionnaire on their attitudes and practices regarding diabetes. The same questionnaires were provided at follow-up to the intervention and control group.

### 2.3. The Intervention

Figure 1 describes the logic model of the study. Both opportunities and challenges are presented to the PHC system in rural China and the health care professionals in PHC. As described in the introduction, the new round of health care reform and the coming healthy China 2030 blueprint enhance the importance of the PHC system, especially for chronic disease care. A number of strategies have been implemented in order to improve PHC. Among these strategies, integrated care aims to strengthen PHC by reinforcing the cooperation with the hospital health care system. However, health care professionals in PHC in rural China commonly have a low level of training [12]. The intervention for health care professionals consisted of three main measures: team communication, regular meetings, and professional skills training sessions. The team communication and regular meetings, which were important components of the integrated care strategy, aimed to bring together health care professionals in the hospital health care system and those in the PHC system at work and to improve the diabetes service to patients. The professional skills training sessions were designed to improve the professional skills and knowledge for those who work in PHC in rural China. As a result, health care professionals were expected to have increased knowledge on diabetes and provide more types of diabetes care service than before. After the intervention, the participants’ attitudes and practices regarding the intervention were asked to assess the effects of team communication and regular meetings. Participants’ professional knowledge was tested and evaluated by a knowledge questionnaire, and questions were asked about the types of diabetes services they were able to provide.

The design of the intervention focused on peer development and peer education delivery. The intervention was conducted from November 2015 to November 2016 by service teams assembled by the county-level HFPC in the intervention areas, which consisted of doctors, nurses, public health physicians, and diabetes specialists, from all three levels of rural health care institutions (county level hospitals, township health care centers, and village clinics). Health care professionals in PHC in the intervention areas received professional skills training sessions driven by the diabetes specialists from county-level hospitals and team communication regarding patient case and participated in regular meetings which aimed to discuss team work progress. In addition, the local HFPC performed technical checks to inspect prevention and treatment plans and performance appraisals to improve the incentive system and encourage professionals to actively participate in the study. Patients in the intervention group received health education lectures, periodical follow-up interviews with an annual physical examination, and special medical services (including helping patients with medical treatment, transfer treatment, return visit, and clinical care). Routine services were provided as usual to those who were in the control areas.

The intervention for health care professionals was controlled and managed by the local county-level HFPC. Detailed information on the intervention is found in Table 1. The professional skills training sessions focused on the professional knowledge and skills in preventing and treating T2DM, including: 1. Typical symptoms of diabetes, and diagnose criteria [4]; 2. Commonly seen diabetic complications (such as diabetic foot and cardiovascular disease); 3. Different treatment strategies, especially insulin treatment and non-drug treatment; 4. Side effects of different medications; 5. Typical cases and treatment; 6. Medication guide. The professional skills training sessions were held every three months in the township health centers in the intervention areas.

The team communication was implemented every two months and separated into two parts: case analysis and inter-team communication. The case analysis mainly focused on exchanging knowledge and experiences about patients whom the staff met, especially patients who had low compliance or difficulties in controlling blood glucose. Meanwhile, team members discussed and assessed the patients’ medication schedule and treatment plan. The inter-team communication included sharing management strategy, reviewing each other’s work, and sharing each other’s experience. The team communication was implemented every two months.

Regular meetings were about reviewing the team work and discuss future plans and were conducted along with the professional skills training sessions. The diabetes specialists in the county-level hospitals would comment and evaluate their work and make suggestions for their future work. Health care professionals working in the control areas did not have any service teams and followed the routine work plan and incentive system as usual.

Ethical approval to collect data from two questionnaires from health care professionals was obtained by Nanjing Medical University Ethics Committee (2015; #300).

### 2.4. Outcome Measures

A structured questionnaire was used to ask the participants regarding their professional knowledge, at baseline and at follow-up data collection one year later. The questionnaire contained two parts: 13 questions related to diabetes professional knowledge, and 8 questions related to services which the health care professionals provided to the patients (see Appendix A
Table A1). A questionnaire on attitudes and practices regarding the intervention was also used and included four sections related to health care professionals’ perspectives on the intervention, including conducting transfer treatment, evaluating cooperation with higher level hospitals, general perspective about PHC institutions, and evaluating factors which affected the integrated care service. Both questionnaires were pilot-tested before use. The Cronbach alpha for the 13 questions related to diabetes professional knowledge and for the 8 questions related to services which the health care professionals provided to the patients was 0.71 and 0.82, respectively. The Cronbach alpha for the questionnaire on attitudes and practices regarding the intervention was 0.84.

For the present study, we extracted three questions about the participants’ attitudes to change and six questions related to the participants’ perspective on their practice before and after the intervention (see Appendix A
Table A2). Each of these questions had five response levels: from the worst to the best.

Participants’ socio-demographic characteristics were collected concerning age (in years), working experience (in years), sex (male vs. female), and medical education level (low vs. high). Participants with technical school degree or junior medical college degree were classified as having low medical education; those with medical college degree or higher degree as having high medical education.

### 2.5. Statistical Analysis

SPSS version 22.0 [27] and Stata 11.0 [28] were employed to analyze the data, using independent t-test and Pearson’s χ2-test to study the difference in sociodemographic characteristics between the intervention and the control group, at baseline data collection. The knowledge score was calculated as a sum of correct answers to each of the questions, and the total score was 13 points. For questions related to attitudes and practice change, one to five points were given to each question, one point representing the worst, and five points representing the best. Participants obtained an attitude score (full score: 15 points) and a practice score (full score: 30 points). To test for differences in outcomes between the intervention and the control group before and after the intervention regarding the knowledge score, attitude score, and practice score, the difference-in-difference model (DID) was adopted, which is commonly used to estimate the effect of a certain intervention by comparing the difference in outcomes over time between an intervention group and a control group [29]. The difference in the proportion of respondents providing different services between the intervention group and the control group after the intervention was analyzed. All other statistical tests were carried out at 5% significance level. 

## 3. Results

### 3.1. Socio-Demographic Characteristics

At the baseline data collection in 2015, 241 health care professionals were selected (132 in the intervention group and 109 in the control group) from 18 township health centers and 55 village clinics. We received 235 knowledge questionnaires and 241 questionnaires on attitudes and practices. At the follow-up data collection in 2016, four (3.0%) participants in the intervention group and five (4.6%) participants in the control group were lost to follow-up. Among those, seven were attending conferences, and the other two were not at their institutions.

Table 2 shows the socio-demographic characteristics as well as the number of years of working experience of participating health care professionals at baseline data collection. The mean age of all participants was 39.8 years, and the mean working experience was 18.4 years. Most of the participants were male (61.4%), and the majority had a low medical educational level (73.4%). There was no statistically significant difference between the intervention and the control group regarding socio-demographic characteristics and working experience, at baseline data collection.

### 3.2. Difference-in-Difference Analysis

Table 3 shows the results of the DID analysis for knowledge, attitudes, and practice scores, comparing the intervention and control group before and after the intervention. At baseline data collection, the practice score and the attitudes score were higher in the control group than in the intervention group, while the knowledge score was slightly higher in the intervention group. However, none of these differences was statistically significant. At follow-up one year after the intervention, participants in the intervention group improved significantly in all three scores, while participants from the control group improved slightly (not statistically significant). The knowledge score, the practice score, and the attitudes score were significantly higher in the intervention group than in the control group. The DID analysis showed that the knowledge score increased significantly more in the intervention group than in the control group after the intervention. Participants in the intervention group also improved significantly more in their practice score and attitudes score than those in the control group.

### 3.3. Proportion of Participants Able to Provide Different Types of Diabetes Services

Table 4 shows the proportion of participants able to provide different types of diabetes services among health care professionals in the intervention and control group. At baseline, there was no significant difference between the intervention group and the control group. Few (less than 30%) health care professionals were able to provide T2DM complication treatment and T2DM emergency service, both in the intervention and in the control group. At follow-up, the proportion of participants being able to provide all eight types of diabetes service in the intervention group increased, compared to the proportion at baseline. There was a substantial and statistically significant higher proportion of participants able to provide services in the intervention group than in the control group, for all types of services, except T2DM emergency treatment. In the intervention group, there were improvements ranging from 10 to 20 percentage points in the proportion of participants providing specific diabetes services: diabetes classification, insulin treatment, oral hypoglycemic agents, early control for T2DM, T2DM complication treatment, and T2DM non-drug treatment. In spite of the increase in the proportion of participants who could provide T2DM complication treatment and T2DM emergency treatment in both two groups, this proportion remained below 50%.

A stratified analysis by the participants’ workplace (township health centers vs. village clinics) and medical educational level (low vs. high) showed similar results as the overall analysis (data not shown).

## 4. Discussion

The socio-demographic characteristics show that the majority of health care professionals had a low-level medical education, which is in line with other studies [12,30,31]. However, the professional knowledge improved significantly in the intervention group, indicating that the professional skills training sessions positively affected the participating health care professionals. Another possible explanation for the improvement is that the design of the knowledge questions was totally based on the training sessions. Therefore, it might have been easier for those who attended the training sessions to get a higher score, which might not reflect their real ability. However, health care professionals in the intervention group were able to provide more types of services than in the control group. Nevertheless, the T2DM emergency service remained unavailable in many PHC institutions in both intervention and control areas, which still needs to be addressed. The improved practice score and attitude score indicate that the participating health care professionals in the intervention group had more positive perspectives than those in the control group regarding cooperation with county-level hospitals, including transfer treatment, communication, and using the new information delivery system.

Patients with common and chronic diseases including T2DM have been encouraged by the National Health Commission since 2015 to use PHC institutions which act as gatekeepers for referral to secondary/tertiary care. This policy aimed to enroll at least 40% of patients with chronic disease in PHC institutions by the end of 2017 [32]. In the present study, we do not have information regarding how the patients in the intervention and control group were actually seeking care, beyond their main contact with PHC institutions.

An intervention in another area in rural China showed a similar result as the present study [24]. A training session was implemented for PHC doctors for 18 months, and the results indicated an improvement in professional skills and increasing ability to provide better quality of diabetes care. Peer education for health care professionals is commonly used in chronic diseases care in many countries [16,33]. The design of the education training session in the present study is similar to that of peer education training. However, unlike the existing studies, the present study emphasized the collaboration between county-level hospitals and PHC institutions and is the only one to our knowledge which combined education sessions and cooperation with hospital-level professional [22,23,24]. Some studies with a similar intervention for diabetes professionals were also conducted in other countries [16,17,18,19,20,21]. Unlike the present study, most of those interventions included the role of peer educators, defined as “community workers who work almost exclusively in community settings. They often serve as connectors between health care consumers and providers to promote health among groups that traditionally lack access to adequate health care” [16]. In the current study, health care professionals in PHC received professional peer education from diabetes specialists in county-level hospitals.

There are several limitations in the present study. The study was not a randomized control trial but was carried out in a real-life setting. As the intervention was implemented in cooperation with county-level HFPC, the implementation may have varied between the counties. Other aspects to be investigated are health care professionals’ personal income and future career plans after the intervention. Only one follow-up data collection was analyzed, but this follow-up period was longer than those evaluated in previous studies [22,23,24]. Longer-term assessment is also recommended.

Nevertheless, the intervention still achieved positive results such as improving professional knowledge and the ability to provide diabetes services, reflected also in the positive impact on patients with T2DM, showing improved fasting blood glucose level and health-related quality of life measured by EQ-5D-3L questionnaire [34]. Hence, an educational intervention to strengthen PHC and reinforce the cooperation between PHC and the hospital health care system, supported by team communication, regular meetings, and professional skills training sessions, could be one way to realize the ambition to shift the care of chronic diseases from hospitals to PHC and improve the care of patients in rural China.

## 5. Conclusions

The increased collaboration between hospitals and primary care services, supported by an educational intervention among health care professionals in PHC, had a positive impact on the professional diabetes skills and knowledge and on the types of services professionals in the intervention group were able to provide, compared with those in the control group, at one-year follow-up. Meanwhile, the attitudes and practices of the intervention group (including cooperation with higher level hospitals, transfer treatment, and information delivery system) also improved, compared with those of the control group. Considering the positive impact of the intervention on patients with T2DM, who showed an improvement in fasting blood glucose, health-related quality of life, and diabetes knowledge, this type of intervention could be tested in other settings of rural China.

## Figures and Tables

**Figure 1 ijerph-17-02076-f001:**
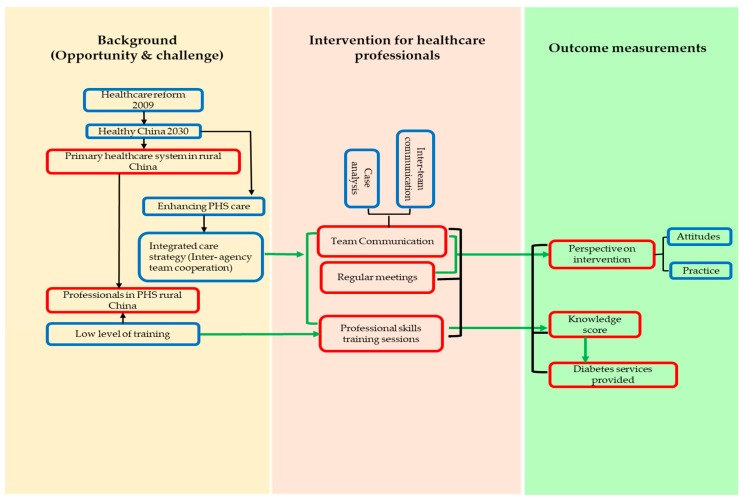
The logic model of the study.

**Table 1 ijerph-17-02076-t001:** Information on the intervention.

	Frequency	Duration	Content	Implementation
**Professional skills training session**	Every three months	2 hours on average	Typical symptoms of diabetes, and diagnose criteria	The professional skills training sessions focused on the professional knowledge and skills in preventing and treating T2DM and was led by the diabetes specialists from county-level hospitals.
			Commonly seen diabetic complications	
			Different treatment strategies, especially insulin treatment and non-drug treatment	
			Side effects of different medications	
			Typical cases and treatment	
			Medication guide	
**Team communication**	Every two months	2 hours on average	Case analysis (Inner-team communication)	Exchange knowledge and experiences on the cases which the participants met.
			Inter-team communication	Share management strategy and experience and review each other’s work
**Regular meeting**	Every three months	1.5 hours on average	Review the team work and discuss future plans	It was conducted along with the professional skills training sessions. The diabetes specialists in the county-level hospitals would evaluate their work and make suggestions for their future work.

**Table 2 ijerph-17-02076-t002:** Socio-demographic characteristics of the participants.

	Intervention (n = 132)		Control (n = 109)		*p*
	n	%	n	%	
**Age**	39.6		40.0		0.777
**Working years**	18.4		18.4		0.995
**Sex**					
Male	84	64.1	64	58.7	0.391
Female	47	35.9	45	41.3	
**Medical educational level**					
Low medical educational level	100	76.9	77	70.6	0.270
High medical educational level	30	23.1	32	29.4	

**Table 3 ijerph-17-02076-t003:** Difference-in-difference analysis (median and interquartile range) of knowledge score, practice score, and attitudes score.

	Baseline							Follow-up							DID	*p*
	Intervention			Control				Intervention			Control					
	Median	25% IQR^*^	75% IQR*	Median	25% IQR*	75% IQR*	*p*	Median	25% IQR*	75% IQR*	Median	25% IQR*	75% IQR*	*p*		
Knowledge score	**8.0**	5.0	9.0	8.0	5.0	10.0	0.961	13.0	11.0	13.0	9.0	4.5	11.0	**< 0.001**	3.65	**< 0.001**
Practice score	14.0	12.0	15.0	14.0	12.5	16.0	0.139	20.0	17.0	23.0	15.5	13.3	17.0	**< 0.001**	5.33	**< 0.001**
Attitudes score	9.0	8.0	10.0	9.0	8.0	10.0	0.258	11.0	9.0	12.0	9.0	9.0	10.0	**< 0.001**	1.71	**< 0.001**

*IQR= Interquartile range. Boldface indicates statistical significance (*p* < 0.05).

**Table 4 ijerph-17-02076-t004:** Difference in the proportion of participants (%) able to provide different types of diabetes service. T2DM: Type 2 Diabetes Mellitus.

	Baseline		Follow-up		Difference	95% CI	*p*
	Intervention	Control	Intervention	Control			
Diabetes diagnose	73.3	73.0	86.3	75.2	11.0	(0.9, 21.1)	**0.032 **
Diabetes classification	59.2	59.1	83.2	61.4	21.8	(10.4, 33.2)	**< 0.001**
Insulin treatment	54.2	53.9	79.4	58.0	21.4	(9.5, 33.3)	**< 0.001**
Oral hypoglycemic agents	76.7	76.5	92.4	79.2	13.2	(4.3, 22.0)	**0.003 **
Early control for T2DM	78.3	78.3	93.9	80.2	13.7	(5.2, 22.1)	**0.001 **
T2DM complication treatment	24.2	24.3	48.9	29.7	19.2	(6.4, 31.9)	**0.003 **
T2DM non-drug therapy	71.7	72.2	92.4	75.2	17.1	(7.8, 26.4)	**< 0.001**
T2DM emergency treatment	29.2	29.6	49.6	37.6	12.0	(-0.9, 24.9)	0.068

Boldface indicates statistical significance (*p* < 0.05).

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
