# Peer review of "Shifting the Care of Type 2 Diabetes Mellitus from Hospital to Primary Health Care Institutions through an Educational Intervention for Health Care Professionals: An Example from Rural China"

_ijerph, 2020, doi:10.3390/ijerph17062076_

Round 1

Reviewer 1 Report

Thank you for submitting your article on Shifting the Care of Type 2 Diabetes Mellitus from Hospital to Primary Health Care Institutions Through an Educational Intervention for Health Care Professionals: an Example from Rural China

I have made the following recommendations to improve the manuscript.  In my opinion the scientific quality of the report writing needs to be tidied up, so that the paper flows and reads more academically. We know that if we provide health professionals with guidance, advice, support and training then practice improves, but what is it that is unique in this intervention here? – perhaps you should elaborate on the peer- delivery model referred to in the discussion??

Please provide further commentary on HOW the intervention was designed, and what was delivered (make this clearer).  Please provide information on the questionnaire measures, specially if you have made these up how these have been piloted, and the relevant reliability statistics (e.g. Cronbach’s alpha etc).

Specific comments:

would be useful to put into a global T2D context

Please capitalise the first word in all abbreviations throughout.  E.g. line 18 Type 2 Diabetes Mellitus  (T2DM) and Primary Health Care (PHC) professionals

Line 34: Type 2 diabetes mellitus (T2DM) is becoming a threat to people’s health in China—and around the world- it would be useful to put this in context globally. As the increase in T2D is not unique to China.

Line 38- thee previous diabetes care strategy- what year- assuming you are referring specifically to china’s strategy.  Is this pre 2009?   As line 42 refers to “In 2009, China launched a”,  this is 11 years ago, is there nothing since, can you cite more current strategy, policy from china?

Line 77- “One county was randomly selected from each part.”  How randomly?

Line 84: can you provide further information: 18 township health centres (9 intervention, 9 control) were part of the study.

-- representing an area of xx population? Or health population characteristics across these areas were….?

Line 90 “Two questionnaires were provided to them – a professional knowledge questionnaire and a questionnaire on their attitudes and practices regarding diabetes. The same questionnaires were provided at follow-up, to the intervention and control group”

Are these validated questionnaires, please provide reference, what is the validity/reliability of the questionnaires used in this context.  Please provide appropriate citations.  If these were newly formed questionnaires please explain how these were developed, and include their reliability stats, and include them within supplementary material.

When was follow-up- what is the timescale here?

Please clarify the actual sentence: “However, health care professionals in PHC in rural China commonly have a low level of training [8].”  Low level of training in x xx x

“The intervention for health care professionals consisted of three main measures: team communication, regular meetings, and professional skills training sessions. The team communication  and regular meetings, which were important components of the integrated care strategy, aimed to  bring health care professionals in the hospital health care system and in the PHC system to work together, and to improve the diabetes service to patients. The professional skills training sessions were designed to improve the professional skills and knowledge for those who work in the PHC in rural China”

[Please provide further information on the design and conduct of the intervention.  In terms of design have you considered a theoretically informed, evidence-based informed, person-based design towards the intervention.  What did the three elements of the intervention actually consist of, so within the team communication- regarding diabetes what examples of communication (messages) were promoted and how, for regular meetings what did a typical agenda look like or what were typical discussion points, for the professional skills training session what were the topics of training? How long, at what level? Etc??  “health education lectures” on what?]     ok I note that this information is in the manuscript however I feel that you could present the content of the intervention better, perhaps insert components into a table or figure to explain.

A questionnaire was used to interview the participants regarding their professional knowledge

I dislike you using the word interview here because you haven’t conducted a qualitative interview, rather participants completed questionnaires regarding their professional knowledge at baselines…..

You had a good retention rate one year post intervention

Line 205-206 “participants in the intervention group improved more in all three scores, while

Participants from the control group improved slightly”- state if significantly different or not

Discussion :  Peer education for health care professionals is commonly used in chronic diseases care in many countries [11,27]. The design of the education training session in the present study is similar to peer education. However, unlike the existing studies, the present study is the only one which combined education sessions and cooperation with the hospital level [17–19].

Ok fine, in the methods you do state who delivers the professional skills training for example, but you haven’t stated that the design of the intervention has focused on peer- development and education delivery- perhaps you would want to improve this aspect of the methods

Line 256 “Other aspects of the intervention studies should investigate the health care professionals’ personal income and future career plan under the intervention.”  In what context is this relevant, please explain.

“up period is longer than previous studies [17–19]. Longer-term assessment is also recommended.”  Longer-term to when?? How long? What other variables could be measured, such as acknowledging that future research to investigate if the health professionals increased knowledge resulted in improvements to actual diabetes care for patients (patients diabetes outcomes for example?).

Reviewer 2 Report

I revised the paper entitled "Shifting the Care of Type 2 Diabetes Mellitus from 2 Hospital to Primary Health Care Institutions Through 3 an Educational Intervention for Health Care 4 Professionals: an Example from Rural China" that aimed to evaluate the impact of an educational intervention on diabetes primary care health care professionals. The topic is relevant and the paper can be interesting for the field. Despite the relevance and the interest of this research, it has some limitations:

  • Sample size estimation was not reported;
  • Models should take into account years of exeriences and the workplace of the professional. hierarchical models are more appropriare in these cases;
  • Questionnaire on knowledges and attitudes are not validated. Cronbach's alpha should be reported;
  • Scores are discrete variables so they should be reported as median and interquartile range;
  • Discussion section is too short. Authors should report comparison with previous similar studies and comparison with studies from other countries. In addition Authors should discuss differences in hosptil setting and in diabetes care among centers involved in the study;
  • In my opinion, Authors should plan an evaluation of patients outcomes after the intervention. were Patients admissions due to diabetes complications decreased? is this intervention useful in terms of patients outcomes?
  • About most common diabetes complications, the intervention focused on all type of complications? any mention of diabetic foot? It is a common complication that can lead to amputation if it is not properly managed in primary care.

Round 2

Reviewer 1 Report

Thank you for systematically addressing my feedback, for which you have mostly addressed. 

Reviewer 2 Report

Authors modified all point raised accordingly with my indications. In my opinion the paper is now acceptable for publication.